# Identification of SLC3A2 as a Potential Therapeutic Target of Osteoarthritis Involved in Ferroptosis by Integrating Bioinformatics, Clinical Factors and Experiments

**DOI:** 10.3390/cells11213430

**Published:** 2022-10-30

**Authors:** Hailong Liu, Zengfa Deng, Baoxi Yu, Hui Liu, Zhijian Yang, Anyu Zeng, Ming Fu

**Affiliations:** 1Department of Joint Surgery, The First Affiliated Hospital of Sun Yat-sen University, Guangzhou 510080, China; 2Guangdong Provincial Key Laboratory of Orthopedics and Traumatology, The First Affiliated Hospital of Sun Yat-sen University, Guangzhou 510080, China; 3Department of Ultrasound in Medicine, The People’s Hospital of PingYi County, Linyi 273399, China

**Keywords:** osteoarthritis, cartilage, ferroptosis, GEO

## Abstract

Osteoarthritis (OA) is a type of arthritis that causes joint pain and limited mobility. In recent years, some studies have shown that the pathological process of OA chondrocytes is related to ferroptosis. Our study aims to identify and validate differentially expressed ferroptosis-related genes (DEFRGs) in OA chondrocytes and to investigate the potential molecular mechanisms. RNA-sequencing and microarray datasets were downloaded from Gene Expression Omnibus (GEO) data repository. Differentially expressed genes (DEGs) were screened by four methods: limma-voom, edgeR, DESeq2, and Wilcoxon rank-sum test. Weighted correlation network analysis (WGCNA), protein-protein interactions (PPI), and cytoHubba of Cytoscape were applied to identify hub genes. Clinical OA cartilage specimens were collected for quantitative reverse transcription-polymerase chain reaction (qRT-PCR) analysis, western blotting (WB), histological staining, transmission electron microscopy (TEM), and transfection. Sankey diagram was used to visualize the relationships between the expression level of SLC3A2 in the damaged area and clinical factors. Based on bioinformatics analysis, clinical factors, and experiment validation, SLC3A2 was identified as a hub gene. It was down-regulated in OA cartilage compared to normal cartilage (*p <* 0.05). Functional enrichment analysis revealed that SLC3A2 was associated with ferroptosis-related functions. Spearman correlation analysis showed that the expression level of SLC3A2 in the OA cartilage-damaged area was closely related to BMI, obesity grade, and Kellgren-Lawrence grade. Furthermore, in vitro experiments validated that SLC3A2 inhibited ferroptosis and suppressed cartilage degeneration in OA. In summary, we demonstrated that SLC3A2 inhibited ferroptosis and suppressed cartilage degeneration in OA. These findings provide a new idea for the study of the pathogenesis of OA, thus providing new means for the clinical diagnosis and targeted therapy of OA.

## 1. Background

Osteoarthritis (OA) is a type of arthritis that causes joint pain, most commonly in the hip and knee joints. It is characterized by cartilage damage, subchondral bone remodeling, synovium inflammation, joint space narrowing, and osteophyte formation [1]. Worldwide, an estimated more than 240 million persons suffer from OA, and this number is still growing with the combined effects of aging and increasing obesity [2]. The pathogenic factors of OA are complex, such as mechanical stress, joint damage, and genetic and metabolic factors. The death of chondrocytes, including autophagy [3,4], necroptosis [5,6], apoptosis [7,8,9], and pyroptosis [10], has been proven to be involved in OA progression.

Ferroptosis, first proposed by Dixon [11] in 2012, is a novel form of programmed cell death. It has different morphological and biochemical characteristics from traditional forms of cell death, mainly iron deposition and lipid peroxidation [12]. Previous studies have shown that the inactivation of GPX4 [13,14] can promote ferroptosis, and some small molecule compounds can also regulate ferroptosis, namely erastin [15], deferoxamine [16], ferrostatin-1 [17]. More and more evidence shows that ferroptosis is closely related to diseases, such as tumors [18,19], neurodegenerative diseases [20,21], lung injury [22], and kidney injury [13]. However, little research is currently available on the role of ferroptosis in osteoarthritis. Yao et al. [23] demonstrated the existence of ferroptosis in OA chondrocytes for the first time, and Miao et al. [24] deeply studied the mechanism of OA ferroptosis caused by the deletion of GPX4. Based on the above findings, ferroptosis may play an important role in the progression of OA. Therefore, identifying ferroptosis-related genes in OA and then studying their mechanisms is very meaningful for the treatment of OA.

In this era of data explosion, the role of bioinformatics analysis in molecular medicine research is becoming more and more prominent. Microarray [25] and RNA sequencing [26] are two techniques commonly used in transcriptomic research and are often used to identify key hub genes. To identify differentially expressed genes related to ferroptosis, we analyzed GEO datasets with bioinformatics methods (e.g., PPI, WGCNA), screened a key gene, SLC3A2, and finally confirmed the function of this gene by cellular experiments. The findings of this study will provide new perspectives on the diagnosis and treatment of OA.

## 2. Methods

The flowchart of this research is shown in Figure 1.

### 2.1. GEO Datasets Acquisition and Preprocessing

Gene expression profiling of OA was downloaded from Gene Expression Omnibus (GEO) data repository (https://www.ncbi.nlm.nih.gov/geo/ (accessed on 20 April 2022)). Two datasets, including GSE114007 and GSE169077, were used in our study. The basic information of the above GEO datasets is shown in Appendix A. Platform annotation file was used to convert “probe id” to “symbol” in the expression matrix. For multiple probes corresponding to the same gene symbol, we calculated the average value as its expression level.

### 2.2. Ferroptosis-Related Genes Collection

FerrDb is the world’s first database that dedicates to ferroptosis regulators and ferroptosis-disease associations (http://www.zhounan.org/ferrdb/ (accessed on 20 April 2022)). We downloaded three datasets (Driver, Suppressor, and Marker) and obtained 388 genes after removing the duplicated genes. The details of all ferroptosis-related genes are shown in Appendix A.

### 2.3. Screening of Differential Expressed Ferroptosis-Related Genes (DEFRGS)

First, genes with expression greater than 0 in at least 90% of the samples in GSE114007 were selected for differential analysis. R package “sva” was used to correct batch effect due to different platforms. Principal component analysis (PCA) was used to evaluate the effectiveness of removing batch effects. Then we used four methods (DEseq2, edgeR, limma-voom, and Wilcoxon test) to identify the differential expressed genes (DEGs). The select criteria of DEGs were set as adjusted *p*-value < 0.05 and |log_2_ fold change| > 1. Finally, DEFRGS were obtained by intersecting DEGs with the above ferroptosis-related genes.

### 2.4. Functional Enrichment Analysis and PPI Network Analysis of DEFRGs

The “symbol” of DEFRGs was converted into “entrezid” for enrichment analysis by R package “org.Hs.eg.db”. Gene Ontology (GO) and Kyoto Encyclopedia of Genes and Genomes (KEGG) pathway enrichment analysis were conducted by using the R package “clusterProfiler”, and the cutoff of *p*-value and q-value was set as 0.05. R packages “ggpubr”, “ggplot2”, and “Goplot” were used for visualization of the enrichment analysis results. The protein-protein interaction (PPI) network was built by using STRING (version11.5, https://cn.string-db.org/ (accessed on 1 May 2022)) with a confidence score >0.4. Cytoscape (version3.9.1) was used for visualization, and the plug-in cytoHubba was performed to calculate the ranking of DEFRGs. We selected the top 20 genes of 10 algorithms (MCC, MNC, EPC, EcCentricity, Degree, Cluster, Closeness, BottleNeck, Radiality, Betweenness) in cytoHubba. R package “UpSetR” was used to identify the hub genes.

### 2.5. Weighted Correlation Network Analysis

To further screen genes that are highly correlated with OA trait, we selected the median absolute deviation (MAD) top 10,000 genes and constructed a gene co-expression network by using the R package “WGCNA”. The steps of data analysis were carried out according to the official tutorial provided by Peter Langfelder and Steve Horvath, UCLA (https://horvath.genetics.ucla.edu/html/CoexpressionNetwork/Rpackages/WGCNA/Tutorials/ (accessed on 10 May 2022)). The gene co-expression modules were obtained by using the blockwiseModules function with default parameters, apart from the soft threshold power, set to 10 (scale-free R^2^ = 0.86), maxBlockSize was 10,000, minModuleSize was 30, mergeCutHeight was 0.4, and TOMType was unsigned. The correlation of modules with the OA trait was calculated by the Pearson correlation coefficient. The two modules with the highest positive correlation and the highest negative correlation were considered hub modules. The threshold points for hub genes in each hub module were |GS| > 0.2 and |MM| > 0.8.

### 2.6. Analysis of the Diagnostic Performance of Marker Genes

The diagnostic performance of marker genes was assessed by the receiver operating characteristic (ROC) curve and area under the curve (AUC). ROC curve analysis was performed with the R package “pROC” and visualized with “ggplot2”.

### 2.7. Cartilage Samples Collection

Osteoarthritis cartilage samples were obtained from patients (*n* = 6; male: 3, female: 3; mean age: 68 years; range: 62–75 years) who underwent total knee arthroplasty (TKA) surgery because of knee OA. Normal cartilage samples were obtained from patients (*n* = 6; male: 3; female: 3; mean age: 47 years; range: 39–57 years) who underwent lower limb amputation due to trauma without osteoarthritis or rheumatoid arthritis. In addition, cartilage tissue and clinical data of 30 patients who underwent TKA due to knee OA were collected. These cartilage samples were classified as damaged (ICRS = 1–4) and undamaged (ICRS = 0) according to the International Cartilage Repair Society (ICRS) grading system [27]. Sample collection was approved by the Ethical Committee of the First Affiliated Hospital of Sun Yat-Sen University, China (IRB: 2011011).

### 2.8. Isolation and Culture of Primary Chondrocytes

The cartilage was cut into small pieces about 1–2 mm in diameter and then washed with phosphate-buffered saline (PBS, Gibco Life Technologies, Grand Island, NY, USA) more than 2 times. Then, the cartilage pieces were added to Dulbecco’s modified Eagle’s medium/Ham’s F-12 nutrient mixture (DMEM/F12, Gibco Life Technologies, USA), containing 1% penicillin (100 IU/mL) and 1% streptomycin (100 μg/mL), and 5% fetal bovine serum (FBS, Gibco Life Technologies), with additional protease (4 mg/mL medium), at 37 °C for 1.5–2 h. Next, the solution was poured off, and the same medium solution as above was added, except that the protease was replaced by collagenase P (0.25 mg/mL medium), at 37 °C for 6–8 h. Finally, primary chondrocytes were collected and seeded in T75 flasks after filtration using a 70 µm cell strainer (BS-70-XBS; Biosharp, Hefei, China) and centrifugation at 1200 rpm for 3 min. All primary chondrocytes were cultured in a DMEM/F-12 medium containing 5% FBS and 1% penicillin/streptomycin solution.

### 2.9. Histological Staining

Cartilage specimens were fixed in 4% buffered paraformaldehyde (Sigma-Aldrich, St. Louis, MO, USA) and subsequently decalcified with 10% EDTA. After embedding in paraffin wax, they were sectioned to 5 μm thickness in a sagittal plane and then stained with Alcian blue, safranin O solution, and toluidine blue.

### 2.10. Transfection Using SLC3A2 Knockdown (KD) shRNAs

Chondrocytes were cultured in 6-well plates and reached approximately 70–80% confluence and then transfected with SLC3A2 KD shRNAs (Tsingke Biotechnology Co., Beijing, China) using Lipofectamine 3000 (Invitrogen, Waltham, MA, USA) according to the manufacturer’s instructions. Non-specific shRNA controls (shNC) were used as negative controls (NCs). Chondrocytes were collected for qRT-PCR after 24–48 h and for western blotting after 48–72 h.

### 2.11. Transmission Electron Microscopy (TEM)

TEM examination was performed 48 h after cell transfection. Chondrocytes were collected in a 1.5 mL Eppendorf tube and fixed with pre-chilled 2.5% glutaraldehyde (pH 7.4) solution for 2 h at 4 °C. After washing with PBS solution, the specimens were fixed with 1% osmium tetroxide for 1–2 h. Next, the specimens were successively dehydrated through a series of different ethanol concentrations (50%, 70%, 90%, 100%) and 100% acetone. Finally, following infiltration, embedding, sectioning, and staining, the specimens were imaged with a Tecnai G2 Spirit Twin transmission electron microscope (FEI Company, Hillsboro, OR, USA).

### 2.12. Quantitative Reverse Transcription-Polymerase Chain Reaction (qRT-PCR)

Total RNA from primary chondrocytes was extracted using RNA-Quick Purification Kit (ESscience, Shanghai, China) and quantified by NanoDrop spectrophotometer (NanoDrop Technologies, Wilmington, DE, USA). Complementary cDNA was synthesized from total RNA using the Evo M-MLV RT master mix (Accurate Biology, Changsha, China). The qRT-PCR assay was performed using the SYBR Green Pro Taq HS qPCR Kit II with ROXdye (Accurate Biology, China) on QuantStudio Real-Time PCR System (Thermo Fisher Scientific, Waltham, MA, USA). Relative gene expression levels were normalized to GAPDH and calculated using the 2^−ΔΔCt^ method. Three technical replicates were set up for each sample. The primer sequences used in our study were: GAPDH-F:5′-GGAGCGAGATCCCTCCAAAAT-3′, GAPDH-R:5′-GGCTGTTGTCATACTTCTCATGG-3′, SLC3A2-F:5′-CTGGTGCCGTGGTCATAATC-3′, SLC3A2-R:5′-GCTCAGGTAATCGAGACGCC-3′.

### 2.13. Western Blotting

Chondrocytes were lysed with RIPA lysis buffer containing protease inhibitors (Beyotime, Shanghai, China). Proteins (20 μg per lane) were separated by 10% SDS polyacrylamide gels electrophoresis (PAGE) and then transferred to activated polyvinylidene fluoride (PVDF) membranes. After being blocked with 5 % skimmed milk, the PVDF membranes were incubated with primary antibodies overnight at 4 ℃. The primary antibodies against SLC3A2 (1:1000, Proteintech, Cat#15193-1-AP) were MMP13 (1:1000, Servicebio, Cat#GB11247-1), COL2A1 (1:1000; Abcam, Cat##ab188570), and GAPDH (1:5000, Proteintech, Cat#HRP-60004). The blots were then incubated with appropriate secondary antibodies for 2 h at room temperature, after which protein bands were detected with an ECL chemiluminescence kit (EMD Merck Millipore, Darmstadt, Germany) by using a chemiluminescence system (Bio-Rad Laboratories, Hercules, CA, USA).

### 2.14. Statistical Analysis

The statistical analyses were performed using R software (version 4.1.2) or GraphPad Prism (v8.0). The samples for all our experiments were obtained from patients with osteoarthritis and controls. Among them, 6 samples from osteoarthritis and 6 samples from control were used for qRT-PCR and WB experiments. The remaining 30 samples from osteoarthritis were used for qRT-PCR and WB, staining, TEM, and transfection. The “*n*” of each experiment has been marked in the corresponding result legend. The Shapiro-Wilk test was used to assess whether the data were normally distributed. Unpaired Student’s *t*-tests (2 groups) and one-way analysis of variance (ANOVA) (multiple groups) were used for values with normal distribution. For values with non-normal distribution, the Mann–Whitney test (2 groups) and the Kruskal Wallis test (multiple groups) were used. The relationships between SLC3A2 expression and baseline characteristics of 30 patients with knee OA were evaluated using Fisher’s exact test and Spearman’s correlation analysis. R^2^ < 0.16 indicates a low linear correlation; 0.16 ≤ R^2^ < 0.49 indicates a significant correlation; 0.49 ≤ R^2^ < 1 indicates a high linear correlation. *p*< 0.05 was considered statistically significant.

## 3. Results

### 3.1. Data Processing and Identification of DEFRGs

GSE114007, including 18 normal and 20 OA cartilage tissues, was used to identify DEGs. The demographic characteristics of the dataset are shown in Appendix A. PCA was used to evaluate the impact of the different platforms on intra-group data. The results showed that the batch effect caused by different platforms was successfully removed (Figure 2a,b). To improve data comparability and reliability, we next normalized the data using the edgeR TMM method, and CPM values were calculated using normalization factors (Figure 2c,d). Four methods, including edgeR, DESeq2, limma-voom, and Wilcoxon test, were used to identify DEGs. The DEGs obtained by the four methods were slightly different according to the same screening criteria (adjusted *p*-value < 0.05 and |log_2_ fold change| > 1) (Figure 2e). A total of 1291 DEGs were obtained by intersecting the results of the four methods (Figure 2f) and presented in a heatmap (Appendix A). To obtain DEFRGs, we intersected these 1291 DEGs with genes obtained from the FerrDb database and finally obtained 54 ferroptosis-related genes (Appendix A).

### 3.2. GO and KEGG Enrichment Analysis of DEFRGs

To study the potential biological functions and signaling pathways of DEFRGs, we performed GO and KEGG enrichment analysis with the R package “clusterProfiler”. The GO analysis results revealed that the DEFRGs were mostly enriched in response to nutrient levels, cellular response to external stimulus, and response to starvation for biology process (BP); vacuolar membrane, basal part of the cell, and basal plasma membrane for cellular component (CC); and ubiquitin-protein ligase binding, ubiquitin-like protein ligase binding, and carbohydrate transmembrane transporter activity for molecular function (MF) (Figure 3a,b). In the KEGG pathway enrichment analysis, DEFRGs were mainly associated with oxidative stress-related pathways, such as mitophagy-animal, ferroptosis, HIF-1 signaling pathway, and mTOR signaling pathway (Figure 3c,d).

### 3.3. PPI Analysis and Hub Genes of DEFRGs

The STRING database was used to explore potential interactions of DEFRGs, and the results were visualized using Cytoscape (Appendix A). We randomly selected 10 algorithms to calculate the score of each node and selected the top 20 genes of each algorithm score as candidate genes. Then we used the R package “UpSetR” to obtain the common genes of 10 algorithms. Finally, a total of 7 hub genes (PTGS2, CDKN1A, TRIB3, RELA, SLC3A2, CEBPG, TSC22D3) were obtained and shown in Appendix A.

### 3.4. WGCNA Analysis

The filtered expression matrix obtained above was normalized by the edgeR TMM method and converted to counts per million (CPM) values. The analysis process was carried out using the R package “WGCNA”. The sample dendrogram and trait heatmap showed the distribution of sample groups and OA traits (Appendix A). All samples were kept because there were no outliers. The pickSoftThreshold function was used to determine the soft threshold power of the scale-free network. When β was set as 10, the scale-free topological model fitting index (R^2^) reached 0.86, and the scale-free network maintained high mean connectivity (Appendix A). Then we used the blockwiseModules function to identify and merge co-expression modules with the parameters set up as minModuleSize = 30 and mergeCutHeight = 0.4 (Figure 4a). A total of 12 modules were identified in the dataset. Each module contained different gene clusters, and the correlation between modules is shown in Appendix A. We next calculated the correlation between modules and OA traits. The results showed that the cyan module had the highest positive correlation, and the blue module had the highest negative correlation (Figure 4b and Appendix A). We calculated gene-module correlations and gene-trait correlations and screened hub genes according to the criteria of |GS| > 0.2 and |MM| > 0.8. The colored parts in the upper right corners of Figure 4c,d are the hub genes of the two modules.

### 3.5. Identification and Validation of Final Hub Genes

A total of two final hub genes were obtained by intersecting PPI hub genes and WGCNA hub genes (Appendix A). Several previous studies [28,29,30] have confirmed that RELA is involved in activities such as chondrocyte differentiation and is closely related to the pathogenesis of OA. However, whether SLC3A2 is related to the occurrence and development of OA has not been reported in the literature. Therefore, SLC3A2 was selected for subsequent studies. We first validated the expression level and diagnostic efficacy of SLC3A2 in another dataset, GSE169077 (Appendix A), and then performed qRT-PCR and WB verification in normal cartilage and OA cartilage specimens (Figure 5a,c,d). Next, we collected cartilage specimens from 6 patients who underwent TKA and divided them into damaged and undamaged areas. Tissue staining, including alcian blue, safranin O, and toluidine blue, showed that the damaged area was relatively smooth, whereas the undamaged area was cracked (Figure 5e). We further detected the relative expression of SLC3A2 using qRT-PCR and WB. We found that the expression of SLC3A2 was significantly downregulated in the damaged cartilage of patients with knee OA (Figure 5b,f,g).

### 3.6. Associations between Clinicopathological Features and SLC3A2

To further identify the clinical significance of SLC3A2 level in OA, we first detected the relative RNA expression of SLC3A2 in the cartilage of the damaged area and undamaged area in 30 OA patients (Figure 6a) and then used the Sankey diagram to visualize the relationships between the expression level of SLC3A2 in the damaged area and clinical factors, including age, gender, obesity grade, K-L grade (Figure 6b,c). Among the 30 patients with OA, we found that 15 had high SLC3A2 levels, and 15 had low SLC3A2 levels in the damaged area (Table 1). Fisher’s exact test results indicated that low SLC3A2 expression was strongly associated with K-L grade (*p* = 0.008) (Table 2). Spearman correlation analysis results confirmed that low expression of SLC3A2 was significantly associated with Body Mass Index (BMI, *p* = 0.01), obesity gradation (*p* = 0.007), and K-L grade (*p* < 0.001) (Table 3). Therefore, the above results indicate that the expression of SLC3A2 was down-regulated in the damaged area of OA cartilage and is related to clinical factors, including BMI, K-L grade, and obesity grade.

### 3.7. Effects of SLC3A2 Knockdown on OA Cartilage

To confirm that SLC3A2 regulates ferroptosis and degeneration of the OA cartilage, we knocked it down using SLC3A2 shRNAs. We performed qRT-PCR experiments to validate the knockdown efficiency of SLC3A2 KD 01 and 02 (Figure 7a). Thus, we used SLC3A2 KD-01 and 02 in subsequent experiments. Our results revealed that transfection of cells with SLC3A2 KD-01 or 02 significantly reversed the changes of OA and increased the levels of MMP13 but decreased COL2A1 (Figure 7b,c). Morphological changes in mitochondria are one of the most important features of ferroptosis cells, which can be observed by transmission electron microscopy (TEM). Compared with the control group, the ultrastructural analysis showed that mitochondria turned smaller, mitochondrial crista was reduced or disappeared, and rupture of the outer mitochondrial membrane occurred in chondrocytes transfected with SLC3A2 KD-01 or 02 (Figure 7d). To further explore whether the function of SLC3A2 in regulating cartilage degeneration is related to ferroptosis, we added Fer-1, a ferroptosis inhibitor, to the culture medium of knockdown cells. Surprisingly, the addition of Fer-1 reversed the effects of knocking down SLC3A2 (Figure 7e,f). Appendix A demonstrates the role of SLC3A2 in ferroptosis and lipid peroxidation-associated progression of osteoarthritis and the potential association between SLC3A2, ferroptosis, and osteoarthritis in pre-clinical and clinical settings.

## 4. Discussion

OA is a common chronic joint degenerative disease, and the damage of periarticular bone and cartilage is an important factor [1]. Ferroptosis has been proven to occur in OA chondrocytes and aggravate the progression of OA. Therefore, it is of great significance to clarify the function of ferroptosis-related genes in chondrocytes for the treatment of OA. Bioinformatics methods can help identify candidate ferroptosis-related biomarkers. Several previous studies have successfully used bioinformatics to identify potential target genes for osteoarthritis. Xia L et al. [31] identified seven key DEFRGs (ATF3, IL6, CDKN1A, IL1B, EGR1, JUN, and CD44) in the synovial tissue of osteoarthritis. Hu Y et al. [32] screened three hub genes (CX3CR1, MYC, and TLR7) and identified potential drugs for the treatment of OA. Duan ZX et al. [33] not only identified PTHLH as the most relevant gene for OA but also further tested and determined PTHLH levels in the plasma and synovial fluid of knee joints of OA patients. In our study, based on the FerrDb database and GEO dataset, the hub gene SLC3A2 was screened, and its function was verified by a series of cellular experiments.

The three parameterization methods, edgeR, limma, and DESeq2, have long been the main methods for differentially expressed gene analysis studies in transcriptomics. However, a new study [34] shows that the classic Wilcoxon rank-sum test has more reliable FDR control and better performance than these three methods when the sample size per group is greater than 8. Therefore, to improve the accuracy of the analysis results, we simultaneously used four methods to screen DEGs. Based on PPI and WGCNA analysis, we identified two candidate genes and further verified the expression with qRT-PCR. The results showed that the expression of SLC3A2 was consistent with the results of our bioinformatics analysis from GSE114007, and we then further confirmed the differential expression and diagnostic effectiveness of SLC3A2 in another GEO dataset (GSE169077), which made our results more convincing.

SLC3A2, also known as 4F2hc, CD98hc, is an 85 kDa type II transmembrane glycoprotein [35]. It often acts as a chaperone protein and forms a heterodimer with some amino acid transporters (e.g., SLC7A5, SLC7A11) to function on the cell membrane [36]. The Xc^-^ system composed of SLC3A2 and SLC7A11 plays an important role in the regulation of ferroptosis. When it is inhibited, the uptake of cystine is restricted, thereby blocking the synthesis of glutathione (GSH), resulting in decreased cellular antioxidant capacity and ferroptosis [37]. Although SLC7A11 plays a major function as a catalytic subunit in the Xc-system, increasing evidence suggests that SLC3A2 is also important for the regulation of ferroptosis. A study on tumor immunotherapy [38] found that interferon released by CD8^+^ T cells can downregulate the expression of SLC3A2 to promote tumor cell ferroptosis. Jianling Bi et al. [39] found that metadherin increases susceptibility to ferroptosis by inhibiting SLC3A2. Additional research [40] has revealed that m6^A^ reader YTHDC2 contributes to ferroptosis in lung adenocarcinoma by targeting SLC3A2. Although SLC3A2 has not been studied in OA, inhibition of its downstream molecule GPX4 has been shown to promote OA [24]. All the above evidence seemed to support our hypothesis that SLC3A2 deficiency or inhibition may exacerbate OA progression by promoting chondrocyte ferroptosis.

To test the above hypothesis, we first explored the differences in the expression of SLC3A2 between OA and non-OA cartilage tissues. Interestingly, we found that the expression of SLC3A2 was significantly down-regulated in OA and related to certain clinical factors. Therefore, we further investigated the expression of SLC3A2 in undamaged and damaged areas of the cartilage and found that it was closely associated with the Kellgren-Lawrence grade. It is well known that the incidence of OA is related to age, gender, obesity grade, weight, BMI, and Kellgren-Lawrence grade. Hence, we explored the relationship between these clinical factors and the level of SLC3A2 expression. We found that the expression of SLC3A2 was weakly correlated with age, and the difference was not statistically significant. This suggests that SLC3A2 may not necessarily be associated with aging. Nonetheless, the expression of SLC3A2 was significantly negatively correlated with BMI and obesity grade.

In vitro experiments showed that SLC3A2 knockdown promoted OA-related cartilage degeneration, as evidenced by the down-regulation of anabolism and up-regulation of catabolism. In addition, its knockdown promoted chondrocyte ferroptosis, and Fer-1 could reverse the effect of the knockdown of SLC3A2 on chondrocytes, which seemed to confirm our hypothesis that SLC3A2 regulates chondrocyte degeneration through ferroptosis. However, the complex mechanisms involved in the regulation of cartilage degeneration by SLC3A2 require further exploration.

Our study has some limitations. First, we used only one dataset to screen for differential genes. Second, there are few cellular functional experiments. Last, no in vivo experiments were performed to further verify the function.

## 5. Conclusions

In summary, we demonstrated that SLC3A2 inhibited ferroptosis and suppressed cartilage degeneration in OA. To the best of our knowledge, this is the first time that the relationship between ferroptosis and SLC3A2 in OA chondrocytes has been demonstrated. These findings provide a new idea for the study of the pathogenesis of OA, thus providing new means for the clinical diagnosis and targeted therapy of OA.

## Figures and Tables

**Figure 1 cells-11-03430-f001:**
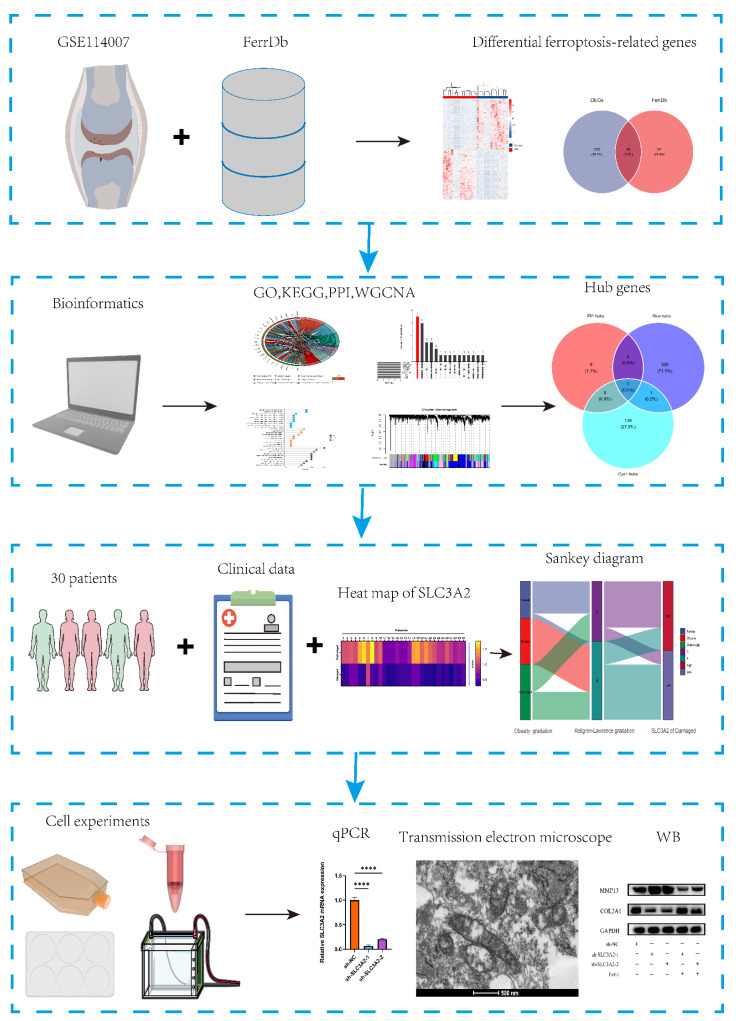
The flowchart of this research. All data were presented as the mean ± SEM: **** *p* < 0.0001.

**Figure 2 cells-11-03430-f002:**
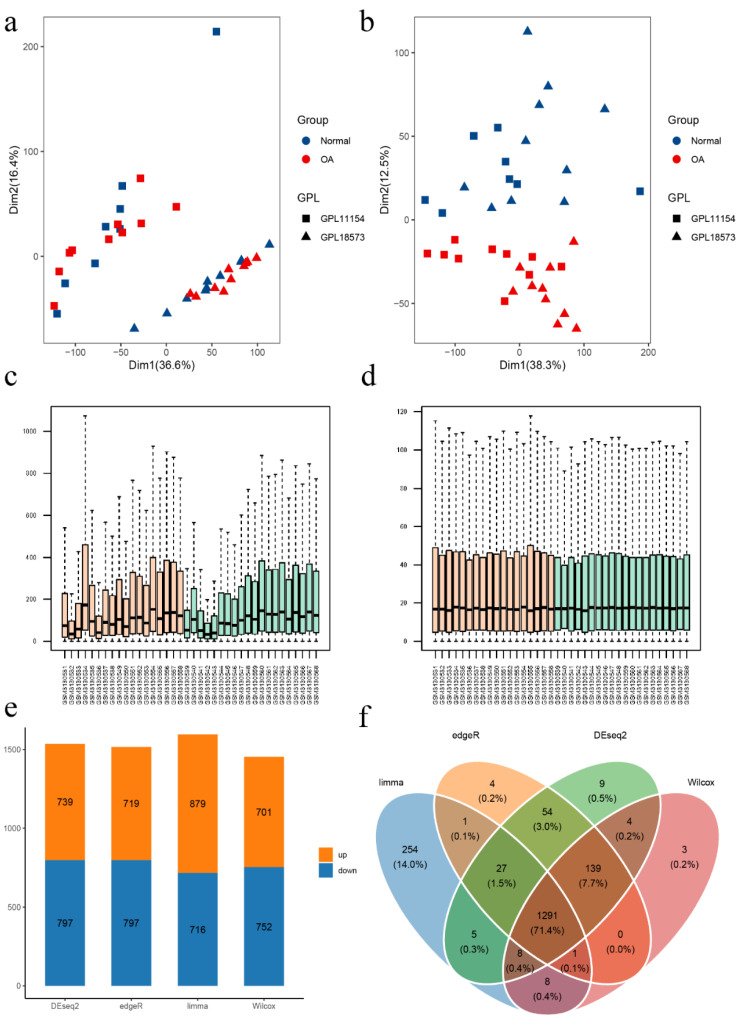
Data preprocessing and identification of DEGs. PCA analyses were performed to visualize the effect before (**a**) and after (**b**) removing the batch effect. Boxplots were used to show data before (**c**) and after (**d**) TMM normalization. (**e**) Differential genes were obtained by four algorithms. (**f**) A Venn diagram was used to obtain the intersection genes of the four methods.

**Figure 3 cells-11-03430-f003:**
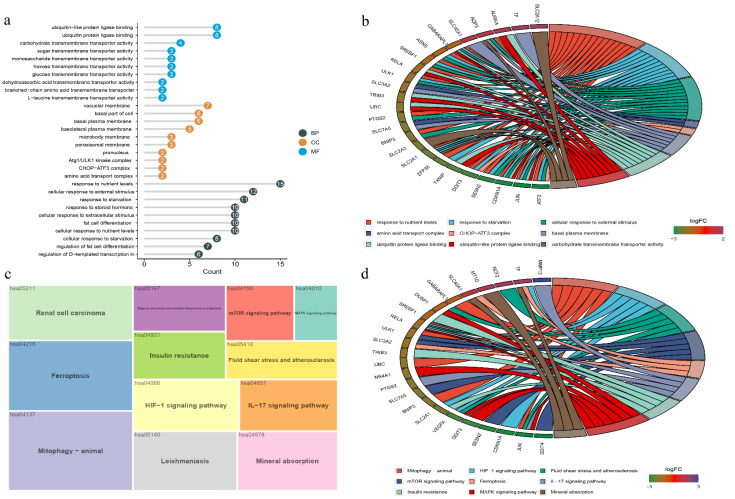
Functional enrichment and PPI of DEFRGs in the normal and OA samples. (**a**,**b**) The results of the GO analysis were displayed in lollipop and circle charts. (**c**,**d**) The results of the KEGG analysis were displayed in tree maps and circle charts.

**Figure 4 cells-11-03430-f004:**
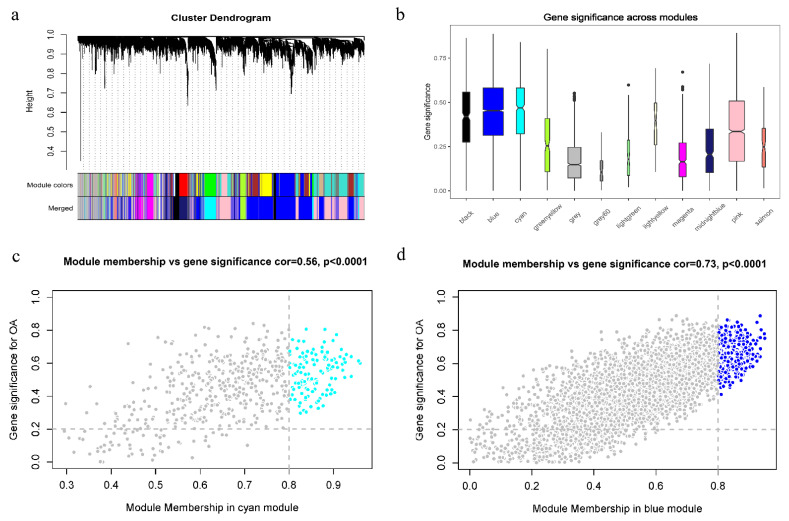
Weighted correlation network analysis. (**a**) Dendrogram of expressed genes in the top 10,000 of MAD clustered based on a dissimilarity measure (1  −  TOM). (**b**) Correlation between OA trait and modules. (**c**,**d**) A scatterplot of Gene Significance (GS) vs. Module Membership (MM) in the module. There is a highly significant correlation between GS and MM in blue and cyan modules.

**Figure 5 cells-11-03430-f005:**
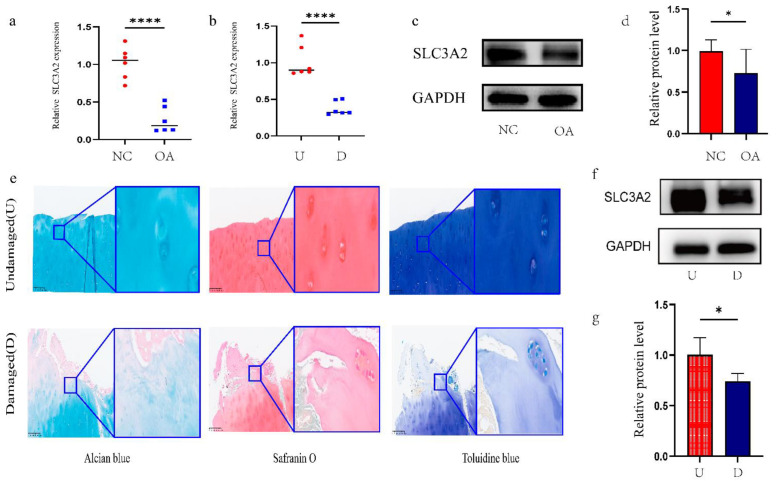
Identification and validation of SLC3A2. (**a**) Validation of differential expression of SLC3A2 in OA and normal samples by qRT-PCR (*n* = 6). (**c**,**d**) Western blotting and quantification analysis in OA and normal samples (*n* = 6). (**e**) Alcian blue, safranin O, and toluidine blue staining. (**b**,**f**,**g**) qRT-PCR and WB analysis in undamaged(U) and damaged (D) areas of OA cartilage(*n* = 6). All data were presented as the mean ± SEM: * *p* < 0.05, and **** *p* < 0.0001.

**Figure 6 cells-11-03430-f006:**
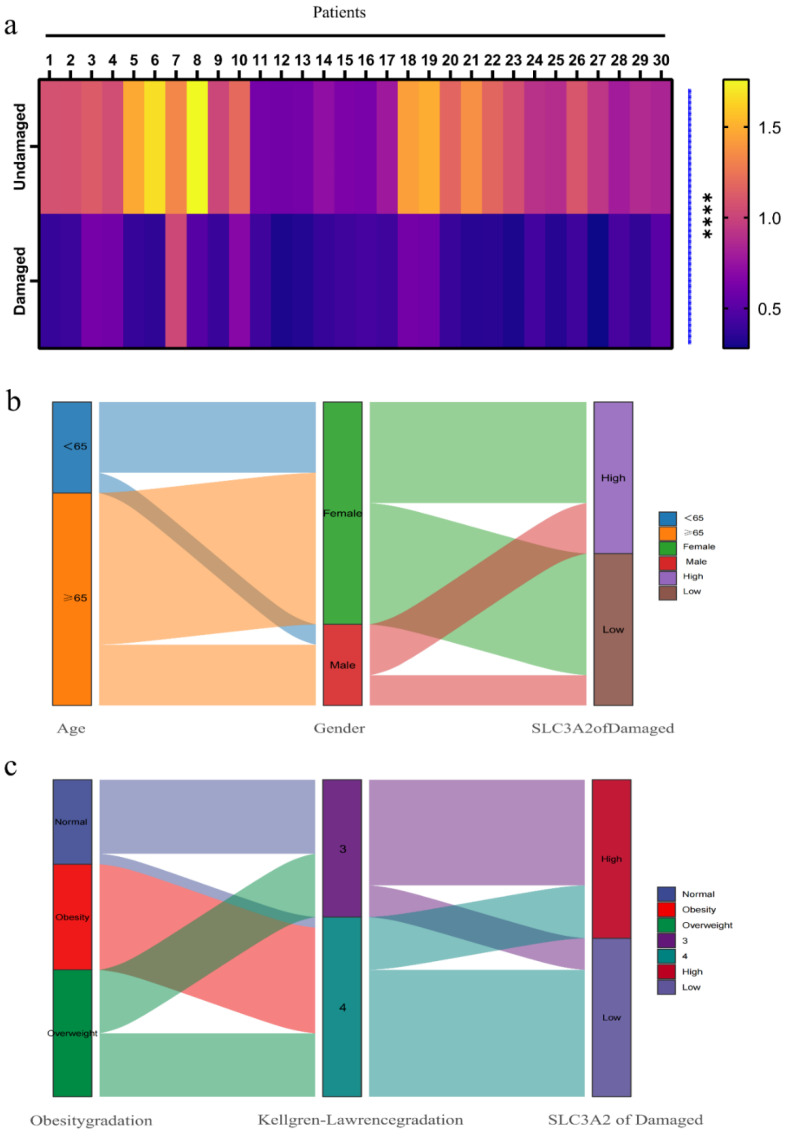
Relationship between SLC3A2 expression and clinical factors. (**a**) Heatmaps of relative RNA expression of SLC3A2 in undamaged (U) and damaged (D) areas (*n* = 30). (**b**,**c**) Sankey diagram presents the relationship between age, gender, obesity grade, Kellgren-Lawrence, and expression of SLC3A2 in the damaged area. All data were expressed as the mean ± SEM: **** *p* < 0.0001.

**Figure 7 cells-11-03430-f007:**
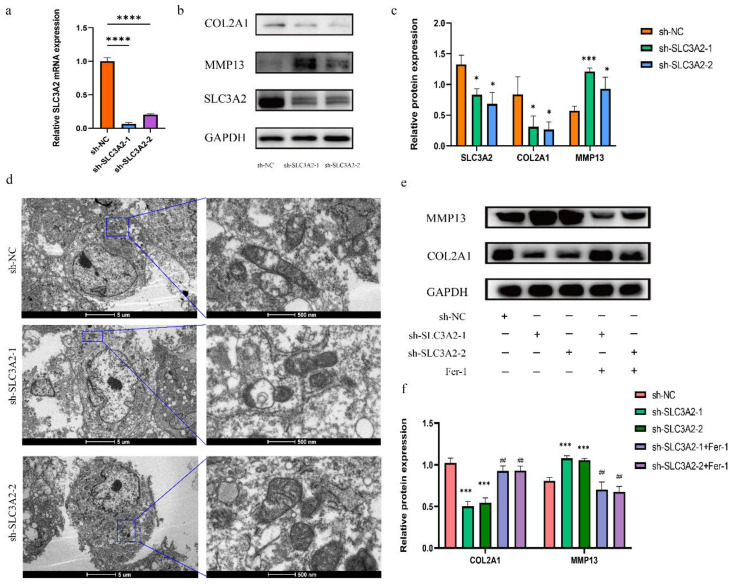
Functional experiments of SLC3A2. (**a**) The efficacy of SLC3A2-specific shRNA was measured by qRT-PCR in chondrocytes. (**b**,**c**) Protein expression of COL2A1, MMP13 in chondrocytes transfected with sh-NC and sh-SLC3A2 analyzed by WB. (**d**) Mitochondrial morphology of chondrocytes transfected with sh-NC and sh-SLC3A2 was observed using transmission electron microscopy (*n* = 3). (**e**,**f**) Fer-1 decreased the MMP13 expression and increased the COL2A1 expression in the transfected chondrocytes. All data were expressed as the mean ± SEM: * *p* < 0.05, *** *p* < 0.001, **** *p* < 0.0001, and ^##^
*p* > 0.05.

**Table 1 cells-11-03430-t001:** Baseline characteristics of patients with knee osteoarthritis (*n* = 30).

Characteristics	Number of Cases (%)
Age (years)	
<65	9 (30)
≥65	21 (70)
Gender	
Male	8 (26.7)
Female	22 (73.3)
Obesity gradation ^1^	
Underweight	0 (0)
Normal	8 (26.7)
Overweight	12 (40)
Obesity	10 (33.3)
Kellgren-Lawrence gradation	
Ⅲ	12 (40)
Ⅳ	18 (60)
Expression of SLC3A2 ^2^	
Low expression	15 (50)
High expression	15 (50)

^1^ Underweight: BMI < 18:5; normal: 18.5 ≤ BMI < 24; overweight: 24 ≤ BMI < 28; obesity: BMI ≥ 28. ^2^ The maximal difference (sixteenth minus fifteenth is 0.0076 in ascending order) near median was used to classify between the low expression or high expression of SLC3A2.

**Table 2 cells-11-03430-t002:** Correlation between SLC3A2 expression and the baseline characteristics of patients with knee osteoarthritis (*n* = 30).

Characteristics	SLC3A2 Expression		*p* Value
	Low	High	
Age (years)			
<65	4	5	>0.999
≥65	11	10	
Gender			
Male	3	5	0.682
Female	12	10	
Obesity gradation			
Underweight	0	0	0.601
Normal	3	5	
Overweight	6	6	
Obesity	6	4	
Kellgren-Lawrence gradation			
Ⅲ	2	10	0.008
Ⅳ	13	5	

*p* values were analyzed using the Fisher’s exact test with *p* < 0.05 as significant.

**Table 3 cells-11-03430-t003:** Spearman correlation analysis between SLC3A2 expression and the baseline characteristics of patients with osteoarthritis (*n* = 30).

Variables	SLC3A2 Expression	
	Spearman	*p* Value
Age	−0.294	0.115
Gender	−0.200	0.289
Weight	−0.234	0.213
Height	0.074	0.697
BMI	−0.462	0.010
Obesity gradation	−0.485	0.007
Kellgren-Lawrence gradation	−0.742	<0.001

Note: *p* < 0.05 considered significant.

## Data Availability

The datasets used during the current study are available in the GEO repository or from the corresponding author on reasonable request.

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
