# Peer review of "Identification of SLC3A2 as a Potential Therapeutic Target of Osteoarthritis Involved in Ferroptosis by Integrating Bioinformatics, Clinical Factors and Experiments"

_cells, 2022, doi:10.3390/cells11213430_

Round 1

Reviewer 1 Report

The authors studied the role of the SLC3A2 gene in the iron-dependent death (ferroptosis) of articular cartilage chondrocytes, thus, in the mechanism of development of osteoarthritis. They screened a library of genes in cartilage tissues obtained from patients (with damaged and undamaged areas) and studied them using RT-PCR, western blot, histological methods, and transfection. They found that SLC3A2 gene inhibited ferroptosis and suppressed osteoarthritis, which represents a new line in the knowledge of osteoarthritis.

This is interesting research, using different methodologies on a substantial number of samples and genes, and with in-depth statistical analysis. The experiments are clearly explained and documented, and the collection of samples was approved by the Ethics Committee. The conclusions are consistent with the results obtained, and a high percentage of references from recent years are included.

However, there are some aspects that should be reviewed:

-      Methods section: it is not clear if all the samples obtained from different patients (with osteoarthritis and controls) are used for all the experiments and therefore the "n" for each group and experiment. These data should be specified in the statistical analysis.

-      The results are explained clearly, but the Figures have too much information and the graphs of each figure are very small and of poor quality, so it cannot be seen clearly. In particular, the histological images (Figures 5e and 7c) need higher quality and larger size since with the actual size and quality they are useless.

-      In the Tables, the horizontal lines (such as the one observed under the "Age" groups in Table 2), would help to better see the data.

Other comments:

-      Check spaces before and after periods (.), commas (,), parentheses, etc. throughout the text and the legend of the figures, since many spaces are missing and, on the other hand, there are many other spaces that must be eliminated.

-      Line 12: the # symbol for the corresponding author is missing.

-      Lines 301-303: the font size seems to be smaller that the rest of the manuscript.

-      Abbreviations (lines 412-422): order them alphabetically for their best location.

-      Reference 15 (lines 470-471) is incomplete: author's last name as well as pages are missing.

Reviewer 2 Report

The manuscript cells-1989594 by Hailong Liu et al. is an interesting work in which the authors, by combining bio-informatics, clinical and experimental data, propose SLC3A2 as a new key player in the progression of the osteoarthritic process. Functional evidences suggest that this new role is mediated through ferroptosis.

The bioinformatics workflow is strong and robust an the clinical data management is also strong.

On the other hand the experimental results are still quite preliminary but promising and they open new research lines to get deep into the mechanism underlying the OA process and suggest new putative therapeutical approaches.

I have some suggestions that, in my opinion, should be addressed by the authors to improve the manuscript before is ready for publication.

11)      The quality, resolution and layout of most of the figures is poor, making difficult to follow the results section. For instance, the quality of the workflow figure must be improved. In figures 2, 3 and 4 some of the schemes could be presented as supplementary figures leaving the important schemes for the main figures. The size and resolution of the text in the figures should also be improved. 

22)      Overall the manuscript is clear and well written with some punctual exceptions (lines 127-128: incomplete sentence; line 183: “blocked”). Please check.

33)      Lines 332-337: The pictures (again with poor resolution) showing the mitochondrial morphological changes could be biased and thay have no biological value unless a proper study is done, with proper statistics and using a specialized image software to measure mitochondrial size and shape, including different TEM preparations and microscopy fields.

Reviewer 3 Report

1.      The authors may include a pictorial presentation of the role of SLC3A2 in ferroptosis and lipid peroxidation-associated progression of osteoarthritis and the potential association between SLC3A2, ferroptosis, and osteoarthritis in pre-clinical and clinical settings.

2.      Discussion section may be strengthened by including a comparison of all results of the current study with those in relevant previous studies listed below:

·             i) Identification and verification of ferroptosis-related genes in the synovial tissue of osteoarthritis using bioinformatics analysis

      https://www.frontiersin.org/articles/10.3389/fmolb.2022.992044/full

·            ii) Identification of Potential Therapeutic Target Genes in Osteoarthritis

      https://www.ncbi.nlm.nih.gov/pmc/articles/PMC9392645/

·            iii) Identification of a potential gene target for osteoarthritis based on bioinformatics analyses

https://josr-online.biomedcentral.com/articles/10.1186/s13018-020-01756-w
